# Recent Changes in the Epidemiology of Group A *Streptococcus* Infections: Observations and Implications

**DOI:** 10.3390/microorganisms13081871

**Published:** 2025-08-11

**Authors:** Susanna Esposito, Marco Masetti, Carolina Calanca, Nicolò Canducci, Sonia Rasmi, Alessandra Fradusco, Nicola Principi

**Affiliations:** 1Pediatric Clinic, Department of Medicine and Surgery, University of Parma, 43126 Parma, Italy; marco.masetti@unipr.it (M.M.); carolina.calanca@unipr.it (C.C.); nicolo.canducci@unipr.it (N.C.); sonia.rasmi@unipr.it (S.R.); alessandra.fradusco@unipr.it (A.F.); 2Università degli Studi di Milano, 20122 Milan, Italy; nicola.principi@unimi.it

**Keywords:** Group A *Streptococcus*, invasive infections, pediatric infectious diseases, antibiotic resistance, COVID-19, vaccine development

## Abstract

*Streptococcus pyogenes* (Group A *Streptococcus*, GAS) is a major human pathogen capable of causing infections ranging from mild pharyngitis and impetigo to severe invasive diseases such as bacteremia, necrotizing fasciitis, and streptococcal toxic shock syndrome (STSS). Historically, the incidence of GAS infections declined during the early antibiotic era but began rising again from the early 2000s, driven partly by the emergence of hyper-virulent strains such as *emm1* and *emm12*. From 2005 onward, significant increases in GAS infections were reported globally, accompanied by rising antibiotic resistance, particularly to macrolides and tetracyclines. During the COVID-19 pandemic, widespread public health measures led to a sharp decline in GAS infections, including invasive cases, but this trend reversed dramatically in late 2022 and 2023, with surges exceeding pre-pandemic levels, notably in children. Recent data implicate factors such as “immunity debt,” viral co-infections, and the spread of virulent clones like *M1UK.* Looking forward, continued surveillance of GAS epidemiology, virulence factors, and resistance patterns is critical. Moreover, the emergence of GAS isolates with reduced susceptibility to beta-lactams underscores the need for vigilance despite the absence of fully resistant strains. The development of an effective vaccine remains an urgent priority to reduce GAS disease burden and prevent severe outcomes. Future research should focus on vaccine development, molecular mechanisms of virulence, and strategies to curb antimicrobial resistance.

## 1. Introduction

*Streptococcus pyogenes*, commonly referred to as Group A *Streptococcus* (GAS), is a highly versatile and globally significant human pathogen. It is capable of asymptomatic colonization, particularly in the pharynx, where it can be detected in up to 20% of healthy individuals [1]. Despite its potential for benign carriage, GAS is responsible for a wide spectrum of diseases that vary greatly in severity, clinical manifestations, and pathogenic mechanisms.

GAS is the leading cause of several common and generally mild infections, including acute pharyngitis, which accounts for a large proportion of sore throat cases in children and adults, and superficial skin infections such as impetigo. However, GAS can also invade sterile sites and cause relatively rare but severe invasive infections (iGAS), such as primary bacteremia, streptococcal toxic shock syndrome (STSS), necrotizing fasciitis, osteomyelitis, and pneumonia. IGAS are often associated with high morbidity and mortality, particularly in vulnerable populations including the very young, the elderly, and individuals with underlying comorbidities [2]. Notably, the clinical presentation of GAS infections may differ substantially between immunocompetent and immunocompromised individuals. In immunocompetent hosts, infections typically remain localized—manifesting as acute pharyngitis, impetigo, or scarlet fever—whereas immunocompromised individuals are at increased risk for fulminant invasive diseases such as bacteremia, necrotizing fasciitis, and streptococcal toxic shock syndrome, often with atypical or attenuated early symptoms [2]. These differences complicate diagnosis and treatment and underscore the need for heightened clinical vigilance and tailored management strategies in high-risk populations.

Beyond acute infections, GAS is also recognized for its role in the development of post-infectious immune-mediated diseases. Acute rheumatic fever (ARF), a sequela of untreated or inadequately treated GAS pharyngitis, can lead to chronic rheumatic heart disease, a significant cause of cardiovascular morbidity worldwide. Similarly, acute post-streptococcal glomerulonephritis can result in transient or, in some cases, permanent renal impairment. These immunological complications add another layer of clinical significance to GAS infections, particularly in regions with limited healthcare resources [2].

The global burden of GAS infections remains substantial. Current estimates indicate approximately 700 million GAS infections per year worldwide, encompassing both pharyngitis and skin infections. Of these, over 650,000 severe iGAS cases occur annually, with an associated 25% case fatality rate, translating into more than 160,000 deaths each year. In addition, rheumatic heart disease (RHD)—a key autoimmune complication following inadequately treated GAS pharyngitis—accounted for an estimated 319,000 deaths in 2015 [3]. This disease burden is compounded by significant economic consequences. In the United States alone, the annual combined costs associated with iGAS disease and acute upper respiratory infections—including direct medical expenses, indirect costs from lost productivity, and the long-term impact of complications—have been estimated at approximately $6.08 billion [4].

These staggering figures highlight the critical importance of prompt diagnosis, effective treatment, and robust preventive strategies in managing GAS infections. The development and implementation of rapid diagnostic techniques for the detection of GAS in clinical specimens have significantly improved the timely initiation of appropriate antimicrobial therapy, thereby reducing transmission and the risk of complications. Concurrently, considerable research efforts are focused on developing safe and effective vaccines targeting GAS, which could dramatically reduce the incidence of both primary infections and their immune-mediated sequelae [5].

In recent years, the epidemiology of GAS infections has shown notable changes, including shifts in the distribution of circulating *emm* types, which encode the highly variable M protein, a major virulence factor and a key target for vaccine development. Moreover, the emergence of GAS strains exhibiting increased virulence and reduced susceptibility to commonly used antibiotics, such as macrolides and clindamycin, has raised significant public health concerns. These trends underscore the potential for changes in disease incidence, severity, and treatment efficacy, necessitating ongoing surveillance and adaptation of clinical management guidelines [5].

Given these developments, it is imperative to reassess the current understanding of GAS epidemiology and its clinical impact. The primary aim of this narrative review is to examine the evolving epidemiological landscape of GAS infections, providing a comprehensive overview of recent trends in disease burden, pathogen characteristics, antimicrobial resistance patterns, and the implications for future prevention and control strategies.

## 2. Methods

To perform this narrative review, we conducted a literature search using PubMed, Scopus, Web A systematic literature search was performed to identify studies investigating the epidemiology, clinical characteristics, and antimicrobial resistance patterns of *Streptococcus pyogenes* (Group A *Streptococcus*, GAS) infections. The electronic databases searched included PubMed/MEDLINE, Scopus, Web of Science, and the Cochrane Library. The search strategy was developed using a combination of Medical Subject Headings (MeSH) terms and free-text keywords related to GAS infections, employing Boolean operators (AND, OR) to optimize sensitivity and specificity. Search terms encompassed various combinations of the following keywords: “group A *Streptococcus*,” “GAS,” “*Streptococcus pyogenes*,” “invasive infections,” “acute pharyngitis,” “skin infections,” “scarlet fever,” “epidemiology,” “COVID-19,” “infants,” and “children.” No language restrictions were applied. The time frame for the search spanned from database inception to April 2025, ensuring comprehensive inclusion of both historical and recent data.

Eligible studies included peer-reviewed original research articles, randomized controlled trials, cohort studies, case–control studies, systematic reviews, meta-analyses, and official reports or guidelines published by recognized health authorities, such as the World Health Organization (WHO), the Centers for Disease Control and Prevention (CDC), the European Medicines Agency (EMA), and the U.S. Food and Drug Administration (FDA). To supplement the electronic search, the reference lists of all relevant articles were manually screened to identify additional studies that might have been missed in the initial search process.

Studies were considered eligible for inclusion if they examined the epidemiology of GAS infections before and after the COVID-19 pandemic, provided data on the severity or clinical spectrum of GAS infections—including both noninvasive and invasive disease presentations—explored antimicrobial resistance patterns in GAS isolates, or analyzed real-world time-related variations in the epidemiology or severity of GAS infections across different regions or populations.

Conversely, studies were excluded if they were narrative reviews, commentaries, editorials, or opinion pieces without original data or systematic methodology. Case reports or small case series involving fewer than five patients were also excluded unless they presented novel epidemiological or antimicrobial resistance information. Additionally, studies were excluded if they lacked sufficient methodological detail to allow an assessment of the validity or quality of the findings.

The process of study selection involved two independent reviewers (MM and CC) who screened titles and abstracts to assess potential relevance. Full-text articles were then retrieved and evaluated in detail to confirm eligibility according to the predefined inclusion and exclusion criteria. Any disagreements regarding study inclusion were resolved through discussion and consensus, with input from a third reviewer (SR) if necessary. Data extracted from the included studies encompassed publication details such as authorship, year of publication, and study location, as well as study design, population characteristics, and outcomes related to GAS epidemiology, disease severity, antimicrobial resistance patterns, and the potential impact of the COVID-19 pandemic on GAS infection dynamics.

## 3. Group A *Streptococcus* (GAS) General Characteristics

GAS is a Gram-positive bacterium belonging to the genus Streptococcus. It is typically classified based on the characteristics of the *emm* gene, which encodes the M protein—a major virulence factor central to the bacterium’s pathogenicity. The M protein plays a critical role in bacterial adhesion, evasion of phagocytosis, and interference with the host immune response. To date, more than 220 distinct *emm* gene variants and their corresponding M protein types have been identified, each exhibiting differences in virulence and tissue tropism [6].

Beyond the M protein, GAS possesses a diverse arsenal of virulence factors that collectively contribute to its pathogenic potential [6,7,8,9,10,11,12,13,14]. Table 1 summarizes key virulence factors of GAS.

The interplay between these bacterial factors and the host immune system substantially influences the severity and clinical manifestations of GAS infections [7]. GAS has mechanisms to degrade host chemokines, DNA, and immunoglobulins, thereby impairing various arms of innate and adaptive immunity. The bacterium secretes numerous extracellular products, including streptolysins S and O (SLS and SLO), which damage host cell membranes and contribute to the characteristic beta-hemolysis observed on blood agar. Additional virulence factors include streptokinase, which promotes fibrinolysis and facilitates bacterial dissemination; hyaluronidase, which breaks down connective tissue and promotes tissue invasion; and cysteine proteinases, particularly SpeB, which degrade a wide range of host proteins and modulate immune responses [8].

Among GAS-secreted proteins, the streptococcal pyrogenic exotoxins (Spes) are particularly significant due to their potent superantigen activity. These exotoxins can bind simultaneously to major histocompatibility complex (MHC) class II molecules on antigen-presenting cells and T cell receptors, triggering polyclonal T cell activation. This excessive immune activation leads to massive cytokine release, which underlies the pathophysiology of severe conditions such as streptococcal toxic shock syndrome (STSS) [9]. Several Spes have been identified, and many are encoded by lysogenic bacteriophages or putative bacteriophage elements integrated into the GAS genome. Notable examples include SpeA, SpeC, SpeG, SpeH, SpeI, SpeJ, SpeK, SpeL, and SpeM, as well as streptococcal superantigen (SSA) and streptococcal mitogenic exotoxin Z (SmeZ). Among these, SpeA and SpeC are the best-characterized superantigens and have been frequently detected in GAS strains isolated from patients with STSS [10,11,12,13].

Specific *emm* types are associated with differing propensities for severe disease, reflecting variations in virulence factor expression. For instance, *emm1* and *emm3* GAS strains have been consistently linked to an increased risk of invasive infections and STSS, whereas strains such as emm4 are generally less associated with severe disease [14].

GAS infections can occur across all age groups but are more prevalent in certain populations. The burden of GAS infections is especially significant in low-income countries, where factors such as overcrowding, poor hygiene, and limited access to healthcare facilitate the spread and severity of infections [15]. However, elevated rates of GAS infections are also observed in indigenous populations within some high-income countries, including Australia and New Zealand. In these communities, a combination of genetic predisposition [16], environmental exposures, and socioeconomic challenges contributes to disproportionately high incidence rates [17].

Seasonality also influences the epidemiology of GAS infections, with the highest incidence typically occurring during winter and early spring. Patterns of disease distribution vary by age and clinical manifestation. Acute pharyngitis and scarlet fever are most commonly diagnosed in children aged 5 to 15 years, whereas impetigo predominantly affects younger children between 2 and 5 years of age [18]. In contrast, iGAS infections are more frequently reported among infants and older adults, particularly those over the age of 65. Sex-specific differences have also been noted, with females more often affected by GAS pharyngitis, while males may exhibit slightly higher rates of iGAS infections [19].

Although noninvasive GAS infections can affect virtually anyone, iGAS infections are more likely to occur in individuals with specific risk factors, including skin injuries, chronic conditions, and immunocompromised status, as shown in several studies highlighting increased susceptibility and worse outcomes in these populations [20,21,22,23]. Viral infections, particularly influenza [24,25] and varicella [26,27,28], are well-established predisposing factors for iGAS disease in children, as these infections compromise mucosal barriers and immune defenses. Furthermore, iGAS infections may arise in the postpartum period, contributing to maternal morbidity and mortality, especially in developing countries where sepsis remains a leading cause of maternal death [29].

## 4. Epidemiology of Group A *Streptococcus* (GAS) Infections

### 4.1. In the First Years of the Antibiotic Era Until 2005

Over the past seventy years, the incidence of GAS infections has undergone significant fluctuations, largely influenced by identifiable factors. During the early decades of the antibiotic era, extending until the late 1980s, the global incidence of GAS infections progressively declined, particularly in industrialized nations, owing to improvements in living conditions and the widespread availability and use of effective antibiotics. From that point until around 2005, GAS infection rates remained relatively stable. However, certain countries reported a modest resurgence in the frequency and severity of GAS infections, including iGAS disease. These unexpected increases were hypothesized to be linked to the circulation of more virulent GAS strains. This hypothesis was supported by evidence indicating that surges in severe infections were frequently associated with specific GAS strains, notably *emm1* GAS [30,31,32].

### 4.2. From 2005 to the Beginning of the COVID-19 Pandemic

Beginning in 2005, reports indicated a steady global rise in GAS infections. In 2011, Hong Kong experienced a significant outbreak of scarlet fever, with case numbers far exceeding the baseline of the previous two decades [33]. Other countries similarly observed gradual increases in GAS infections, which eventually reached levels substantially higher than those reported in earlier periods. Notably, although overall GAS infections increased, the most pronounced rise was seen in severe cases and iGAS infections. In Hong Kong, the 2011 outbreak included a higher proportion of severe scarlet fever cases, some complicated by STSS. Between 2012 and 2015, Ireland recorded iGAS incidence rates approximately two to three times higher than those observed during the preceding eight years [34]. In the United States, the incidence of iGAS increased from 1.04 per 100,000 people in 2008 to 4.76 per 100,000 in 2019 [35]. Similarly, in Australia, iGAS incidence rose from 4.1 per 100,000 in 2008/2009 to 8.3 per 100,000 in 2017/2018 [36].

Microbiological investigations during this period confirmed that the rising burden of iGAS was strongly associated with increased circulation of GAS strains exhibiting enhanced virulence. Isolates recovered from patients with iGAS infections frequently produced superantigens and displayed resistance to commonly used antibiotics. For instance, a study from Hong Kong [37] revealed that most scarlet fever cases were caused by emm12 GAS strains exhibiting resistance to macrolides and tetracyclines and carrying the SSA and SpeC superantigens. In Ireland, iGAS infections were primarily associated with the *emm1*, *emm3*, *emm28*, *emm12*, and *emm89* strains, which produced various superantigens, including SpeA, SpeG, and SpeJ [38]. In the United Kingdom, the rise in iGAS incidence coincided with the emergence of a novel lineage of the *emm1* strain, designated *M1UK*, distinguished by its elevated expression of the scarlet fever toxin SpeA [39].

During this period, the increasing prevalence of antibiotic resistance among GAS isolates was recognized as a significant driver of the spread of virulent strains. Although resistance rates varied by region, substantial increases in GAS resistance to macrolides, lincosamides, tetracyclines, fluoroquinolones, and sulfonamides were documented worldwide. Notably, resistance often clustered in GAS strains linked to severe disease and iGAS. In Taiwan, macrolide resistance in GAS rose from 18.1% to 19.3% between 2000 and 2009, escalating further to 58.4–61.6% in the subsequent decade, with most resistant isolates identified as *emm12* GAS [40]. A study from China [41] documented a striking increase in the prevalence of *emm1* GAS among scarlet fever cases, rising from 3.8% in 2011 to 48.6% in 2014. Nearly all isolates from patients (*n* = 1451) and carriers (*n* = 96) demonstrated resistance to erythromycin (97.5%), clindamycin (97.3%), and tetracycline (95.7%). Tetracycline resistance varied significantly by geography, ranging from about 10% in Australia to over 80% in China. While resistance to fluoroquinolones remained generally lower, clinically significant levels were reported in Hungary (10%) and Japan (14%) [42].

### 4.3. The Period of the COVID-19 Pandemic

The epidemiology of GAS infections underwent marked changes during and following the COVID-19 pandemic. Contrary to the upward trend noted before the pandemic, GAS infections—including iGAS—decreased sharply in the early years of SARS-CoV-2 circulation worldwide. According to data from the Centers for Disease Control and Prevention (CDC), the incidence of iGAS in the United States fell 52%, 74%, and 27% below expected levels in 2020, 2021, and 2022, respectively. Declines were observed across all high-risk populations, with incidence reductions of 73% among children and 33% among adults aged 65 and older [43]. Similarly, data from an Italian tertiary center (Policlinico Agostino Gemelli, Rome) demonstrated a 50% reduction in diagnoses of streptococcal pharyngitis and other GAS infections in 2020, followed by an additional 30% decrease during 2021–2022 compared to pre-pandemic years [44].

This declining trend reversed dramatically during the latter months of 2022 and into early 2023, when a sharp resurgence in GAS infections was documented worldwide. Incidence levels during this period surpassed those observed in pre-pandemic years, with the increase being particularly pronounced for iGAS infections in younger children, though older individuals continued to experience the highest recorded overall incidence rates. A growing proportion of iGAS cases required hospitalization and intensive care unit admission. In France, a study documented a significant rise in severe iGAS infections during the later pandemic period, with longer hospital stays (9.4 vs. 6.4 days; *p* = 0.0007), higher rates of pleural empyema (15% vs. 4%), bone and joint infections (14% vs. 7%; *p* = 0.008), and more complicated ENT infections (22% vs. 7%; *p* = 0.003) [45]. In the UK, by June 2023, the incidence of scarlet fever and iGAS infections exceeded the five-year pre-pandemic average [46]. In the United States, the observed incidence of iGAS during November and December 2022 was 117% higher than expected [45]. At the previously mentioned tertiary center in Rome, the proportion of positive pharyngeal swabs for GAS rose to 13% in 2023, compared to just 2% in 2022 [44].

In the latter part of 2023 and into early 2024, provisional data suggested that GAS incidence had declined again but remained modestly above typical seasonal expectations. In the UK, reports indicated that scarlet fever notifications followed typical seasonal patterns, albeit at the higher end of historical ranges. Meanwhile, iGAS incidence returned to expected levels, predominantly affecting older adults rather than children, in contrast to the spike seen in the immediate post-pandemic period [46].

Both the sharp reduction in GAS infections during the early COVID-19 pandemic and the subsequent resurgence are primarily attributed to public health measures aimed at controlling SARS-CoV-2 transmission. Nonpharmaceutical interventions (NPIs)—including travel restrictions, event cancelations, social distancing, curfews, and lockdowns—not only curtailed the spread of COVID-19 but also significantly reduced the transmission of other respiratory pathogens. Consequently, there was a marked decline in respiratory infections and related healthcare utilization. The effect was particularly profound for nonenveloped viruses such as influenza and respiratory syncytial virus (RSV), whereas enveloped viruses like rhinovirus, enterovirus, human bocavirus, and human adenovirus were less affected due to differences in environmental stability and susceptibility to disinfection [47]. Similar trends were documented for other bacterial pathogens; alongside GAS, there was a substantial decrease in *Streptococcus pneumoniae* infections. Multiple studies reported dramatic declines in invasive pneumococcal diseases (IPDs), including bacteremic pneumonia, during the early pandemic period, particularly among children under five years of age [48,49,50,51,52,53,54].

As NPIs were gradually relaxed in the later stages of the pandemic, previously suppressed pathogens resumed circulation, albeit at varying speeds. Some pathogens quickly returned to pre-pandemic levels, while others showed delayed resurgence, sometimes only reaching baseline levels after three years [47]. Notably, RSV infections exhibited an off-season resurgence [55], occasionally accompanied by shifts in age distribution toward older children [56].

Several factors likely contributed to the increased incidence of respiratory infections during the late pandemic period. Reduced exposure to common pathogens led to gaps in population-level immunity, particularly among infants born during the pandemic and women of childbearing age, resulting in greater susceptibility once pathogen circulation resumed—a phenomenon sometimes described as “immunity debt” [57]. Some studies have hypothesized that children with prior SARS-CoV-2 infections may experience immune dysregulation, which could increase susceptibility to other infections, including invasive disease caused by virulent strains of GAS; however, further research is needed to clarify this potential link [58,59,60]. Concurrent viral infections may also have played a role. Influenza and varicella are well-established risk factors for subsequent GAS infections, including severe invasive cases, likely due to the immunosuppressive effects of viral infections and damage to mucosal barriers, which facilitate bacterial invasion [58].

Supporting this link, a study in Denmark during the 2022–2023 season found that 22% of pediatric iGAS cases were preceded by varicella infection, and 47% followed an upper respiratory viral infection [59]. In England, data from October to November 2022 identified RSV and human metapneumovirus (hMPV) as the most common co-infecting viruses in pediatric GAS cases [60]. Similarly, a Portuguese study conducted between September 2022 and May 2023 observed that varicella and upper respiratory infections preceded iGAS in 24.6% of cases [61]. Further evidence from the Netherlands indicated that between January 2022 and March 2023, 34% of GAS skin and soft tissue infections (SSTIs) in children were attributable to varicella. Additionally, among pediatric and adult iGAS pneumonia or sepsis cases in the same period, 34% (95% CI: 20–49) and 25% (95% CI: 18–32), respectively, were linked to preceding respiratory virus infections, with influenza A contributing the largest share at 17% [62].

While the concepts of immunity debt and increased viral infections undoubtedly contributed to the overall rise in GAS infections, it is highly probable that the resurgence of iGAS was primarily driven by the increased circulation of highly virulent GAS strains already present prior to the pandemic. By late 2022, in the Netherlands, the proportion of *emm1* among iGAS isolates rose from 32% in November to 68% in December, maintaining levels above 50% until March 2023 [63]. Moreover, in this same period, emm1 strains were largely replaced by the M1UK variant [63], which had undergone further evolution, acquiring new virulence determinants [64].

Table 2 shows results from main studies with global incidence data for iGAS, whereas Table 3 summarizes studies on iGAS antimicrobial resistance patterns.

Figure 1 synthesizes major epidemiological milestones, changes in dominant GAS phenotypes, and shifts in incidence patterns, including the impact of the COVID-19 pandemic.

## 5. Future Outlook for Group A *Streptococcus* Infections

Although data collected in 2024 remain provisional, current trends suggest that, barring extraordinary events such as the COVID-19 pandemic, the epidemiology of GAS infections will likely return to patterns observed prior to the pandemic. The overall number of GAS infections is expected to stabilize at levels similar to, or somewhat higher than, those seen before the pandemic, with a modest yet persistent rise in severe cases. This increase may be driven by the continued circulation of highly virulent GAS strains. The situation is further complicated by growing concerns over reduced susceptibility of certain strains to commonly used antibiotics, which can promote the selection and broader dissemination of resistant bacteria [65].

To mitigate the future burden of GAS infections, several critical measures are essential. Foremost among these is the sustained surveillance of GAS epidemiology and ongoing characterization of circulating strains to assess virulence factors and antimicrobial susceptibility profiles. Understanding which strains most frequently cause severe disease is crucial for informing public health interventions and for guiding the development of an effective vaccine. Despite numerous attempts, no vaccine is currently available for GAS, leaving a significant gap in preventive strategies.

Additionally, rigorous monitoring of antimicrobial resistance trends is imperative to ensure appropriate therapeutic choices and to preserve the efficacy of existing treatments. Special attention should be directed toward the emergence of GAS isolates with reduced susceptibility to beta-lactam antibiotics. Although no naturally occurring beta-lactam-resistant GAS strains have been definitively documented, beta-lactams remain the first-line therapy for all GAS infections. Recent reports of isolates with decreased susceptibility to penicillin have raised legitimate concerns regarding the potential evolution of true beta-lactam resistance. Reduced susceptibility has been linked to specific mutations in the penicillin-binding protein 2X (PBP2X), which alter its structure and diminish its affinity for beta-lactam antibiotics, leading to elevated minimal inhibitory concentrations *in vitro* [66,67].

Looking ahead, two areas of particular importance are vaccine development and the advancement of novel antimicrobial strategies—especially in the context of rising antimicrobial resistance and the clinical challenges posed by rapidly progressing infections such as necrotizing soft tissue infections. Despite decades of effort, no licensed vaccine for GAS is currently available. However, recent progress has renewed optimism, with several candidates under preclinical and early clinical investigation. These include multivalent M protein-based vaccines targeting the most prevalent emm types globally, as well as vaccines utilizing conserved antigens such as streptolysin O, C5a peptidase, and Group A Carbohydrate (GAC), which aim to overcome the issue of strain variability and ensure broader protection [68,69]. Additionally, the use of mRNA and nanoparticle platforms is under early-stage evaluation for GAS vaccines [70].

On the therapeutic front, novel antimicrobial approaches are gaining attention as adjuncts or alternatives to conventional antibiotics. These include bacteriophage-derived lysins with specific lytic activity against GAS, engineered antimicrobial peptides with broad-spectrum efficacy, and anti-virulence compounds targeting toxin production or quorum sensing. Such strategies are particularly relevant for managing life-threatening infections like necrotizing fasciitis or streptococcal toxic shock syndrome, where rapid disease progression and emerging resistance can compromise the effectiveness of β-lactams and clindamycin [71,72,73]. Continued research and investment in these innovative avenues will be critical to expanding our clinical armamentarium and improving outcomes in high-risk patients.

## 6. Conclusions

In summary, GAS remains a clinically significant pathogen with the capacity to cause both widespread, mild infections and severe, life-threatening invasive diseases. The shifting epidemiology observed over recent decades underscores the importance of vigilant surveillance to detect emerging virulent strains and track evolving antimicrobial resistance patterns, including potential threats to beta-lactam efficacy. The development of a safe and effective GAS vaccine represents a critical unmet need that could transform the landscape of GAS prevention, particularly in high-risk populations and regions with high disease burden. Future research should prioritize the identification of conserved vaccine targets across diverse emm types, the mechanisms underlying increased virulence and invasiveness, and strategies to mitigate the rise in antimicrobial resistance. Addressing these gaps will be essential to reduce the global burden of GAS disease and to safeguard the effectiveness of current treatment options for years to come.

## Figures and Tables

**Figure 1 microorganisms-13-01871-f001:**
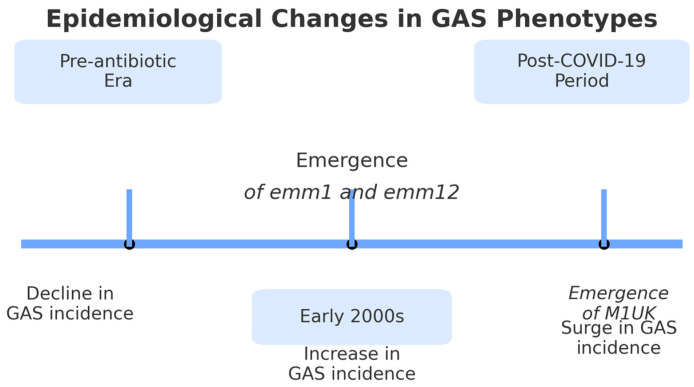
Timeline of epidemiological changes in Group A *Streptococcus* (GAS) phenotypes from the pre-antibiotic era to the post-COVID-19 period.

**Table 1 microorganisms-13-01871-t001:** Key virulence factors of Group A *Streptococcus.*

Virulence Factor	Function	Associated Clinical Impact	Relevant Strains (Examples)
**M Protein (*emm* types)**	Inhibits phagocytosis, promotes adhesion	Linked to invasiveness, tissue tropism	*emm1*, *emm3*, *emm4*, *emm12*, *emm89* [6,14]
**Streptolysin O (SLO)**	Pore-forming cytotoxin damaging host cells	Beta-hemolysis, tissue injury	Broadly distributed [8]
**Streptolysin S (SLS)**	Cytolytic toxin contributing to hemolysis	Tissue damage, immune evasion	Broadly distributed [8]
**Streptokinase**	Converts plasminogen to plasmin, promotes fibrinolysis	Facilitates tissue invasion	Broadly distributed [8]
**Hyaluronidase**	Degrades connective tissue matrix	Promotes spread through tissues	Broadly distributed [8]
**SpeB (cysteine protease)**	Degrades host proteins and immune mediators	Modulates immune response, promotes tissue destruction	Various *emm* types [8]
**Pyrogenic Exotoxins (SpeA, SpeC, etc.)**	Superantigen activity leading to massive cytokine release	Linked to STSS, severe invasive disease	SpeA, SpeC in *emm1*, *emm3*, *M1UK* [9,10,11,12,13]

STSS: Streptococcal toxic shock syndrome.

**Table 2 microorganisms-13-01871-t002:** Main studies with global incidence data for invasive GAS infections.

Country/ Region	Time Period	iGAS Incidence (Per 100,000 Population)	Notable Strains or Findings	References
USA	2008	1.04	Baseline prior to steady increase	[35]
USA	2019	4.76	Rising incidence noted	[35]
Australia	2008/2009	4.1	Rising trends reported	[36]
Australia	2017/2018	8.3	Significant increase	[36]
Ireland	2012–2015	~2–3× higher than 2004–2011	Predominance of *emm1*, *emm3*, *emm28*, *emm12*, and *emm89* strains	[34,38]
Hong Kong	2011	Surge in scarlet fever cases	Linked to the *emm12* strain with toxin acquisition and resistance	[33,37]
Netherlands	Nov–Dec 2022	Increase from 32% to 68% of emm1 among iGAS	Emergence and dominance of the *M1UK* variant	[63]

iGAS, invasive Group A *Streptococcus*.

**Table 3 microorganisms-13-01871-t003:** Antimicrobial resistance patterns in GAS (2000–2024).

Region/ Country	Antibiotic	Resistance Rate (%)	Associated *emm* Types/Notes	References
Taiwan	Macrolides	18.1% (2000–2009) → 58.4–61.6% (2010–2019)	Predominantly *emm12* strains [40]	[40]
China	Macrolides	~97.5%	High resistance among both patients and carriers	[41]
China	Clindamycin	~97.3%	Significant resistance documented	[41]
China	Tetracyclines	~95.7%	Notable regional variability	[41]
Australia	Tetracyclines	~10%	Lower resistance rates observed	[42]
China	Tetracyclines	>80%	Substantially higher than global averages	[42]
Hungary	Quinolones	~10%	Notable but relatively low resistance overall	[42]
Japan	Quinolones	~14%	Emerging resistance trend	[42]

## Data Availability

No new data were created or analyzed in this study.

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
