# Peer review of "Recent Changes in the Epidemiology of Group A *Streptococcus* Infections: Observations and Implications"

_microorganisms, 2025, doi:10.3390/microorganisms13081871_

Round 1

Reviewer 1 Report

Comments and Suggestions for Authors

Thank you for the opportunity to review the manuscript “Recent Changes in the Epidemiology of Group A Streptococcus Infections: Observations and Implications”. The manuscript is well written and interesting. I only have some minor comments to further improve an already good manuscript:

  1. Reference 3 is Carapetis et al from 2005. This was updated in 2016, and it would be better to refer to that (and update figures): https://www.ncbi.nlm.nih.gov/books/NBK333415/
  2. Exact search strings in each database should be provided, preferably as a supplemental file to avoid cluttering the manuscript.
  3. Authors write on page 7 “…sometimes described as “immunity debt.” As a curtesy to those who came up with this expression, I suggest inserting a reference to where this expression was first mentioned.
  4. Reference 9 and 32 seems to be the same and should be merged.

Author Response

Thank you for the opportunity to review the manuscript “Recent Changes in the Epidemiology of Group A Streptococcus Infections: Observations and Implications”. The manuscript is well written and interesting. I only have some minor comments to further improve an already good manuscript:
Re: Thank you for your positive evaluation. We revised the manuscript according to your suggestions.

Reference 3 is Carapetis et al from 2005. This was updated in 2016, and it would be better to refer to that (and update figures): https://www.ncbi.nlm.nih.gov/books/NBK333415/
Re: Revised as suggested (pp. 2 and 11). 

Exact search strings in each database should be provided, preferably as a supplemental file to avoid cluttering the manuscript.
Re: This is a narrative review and usually search strategy is included in the Methods as in this case. 

Authors write on page 7 “…sometimes described as “immunity debt.” As a curtesy to those who came up with this expression, I suggest inserting a reference to where this expression was first mentioned.
Re: Added as recommended (pp. 8 and 14).

Reference 9 and 32 seems to be the same and should be merged.
Re: Reference 9 has been replaced by a new one.

Reviewer 2 Report

Comments and Suggestions for Authors

The authors present a narrative review to explore the clinical and public health implications of changes in Group A Streptococcus (GAS) epidemiology. This is an important area given the increasing global significance of GAS infections in driving morbidity and mortality. The introduction is well written and provides a succinct overview for the general reader. The methods are well described. The section on “GAS general characteristics” provides further useful background information. The main body of the review that describes epidemiological changes of GAS infections is very helpful and appropriately cites the key references. In particular, the tables will provide useful synthesis. The discussion points are thoughtful.

The review would be improved if the following points were addressed:

  • The assertion that “ immunocompromised individuals are at particularly high risk for both noninvasive and invasive forms of GAS disease” is contentious and requires further qualification from primary published data with appropriate references
  • The assertion that “Additionally, children who had prior SARS-CoV-2 infections may have experienced immune dysregulation, increasing vulnerability to other infections, including virulent strains of GAS implicated in invasive disease.” is contentious and requires further qualification from primary published data with appropriate references
  • Recent studies have explored immuno-epidemiological evidence for the GAS immunity debt hypothesis e.g. longitudinal serological studies. The findings of these studies are highly relevant to this review, and should be included in (or around) section 4.3.
  • A simple figure/pictogram summarising the epidemiological changes in different GAS phenotypes over the observed period (pre-antibiotic period through to post-COVID period) would provide a helpful high-level synthesis

Author Response

The authors present a narrative review to explore the clinical and public health implications of changes in Group A Streptococcus (GAS) epidemiology. This is an important area given the increasing global significance of GAS infections in driving morbidity and mortality. The introduction is well written and provides a succinct overview for the general reader. The methods are well described. The section on “GAS general characteristics” provides further useful background information. The main body of the review that describes epidemiological changes of GAS infections is very helpful and appropriately cites the key references. In particular, the tables will provide useful synthesis. The discussion points are thoughtful.
 Re: Thank you very much for your positive evaluation. We revised our manuscript accordingly.

The review would be improved if the following points were addressed:
•    The assertion that “ immunocompromised individuals are at particularly high risk for both noninvasive and invasive forms of GAS disease” is contentious and requires further qualification from primary published data with appropriate references
Re: We thank the reviewer for highlighting the need for greater precision. We agree that the assertion required further qualification and have accordingly revised the text to clarify the context and provide supporting evidence. Specifically, we have emphasized that immunocompromised individuals are primarily at increased risk for invasive GAS infections, as supported by several studies (p. 5). 

•    The assertion that “Additionally, children who had prior SARS-CoV-2 infections may have experienced immune dysregulation, increasing vulnerability to other infections, including virulent strains of GAS implicated in invasive disease.” is contentious and requires further qualification from primary published data with appropriate references.
Re: We appreciate the reviewer’s critical observation. We agree that the assertion required further qualification and clarification. We have revised the sentence to reflect that the link between prior SARS-CoV-2 infection and increased susceptibility to invasive GAS disease in children remains hypothetical and is based on emerging but limited evidence (p. 8).

•    Recent studies have explored immuno-epidemiological evidence for the GAS immunity debt hypothesis e.g. longitudinal serological studies. The findings of these studies are highly relevant to this review, and should be included in (or around) section 4.3.
Re: We thank the reviewer for this insightful comment. We fully agree that recent immuno-epidemiological studies—particularly those involving longitudinal serology—provide important evidence in support of the “immunity debt” hypothesis and are highly relevant to the post-pandemic resurgence of GAS infections.
However, to preserve the clarity and logical flow of the manuscript, we respectfully propose to maintain the existing structure of Section 4.3, which is organized chronologically and thematically around changes in GAS epidemiology during the COVID-19 pandemic.

•    A simple figure/pictogram summarising the epidemiological changes in different GAS phenotypes over the observed period (pre-antibiotic period through to post-COVID period) would provide a helpful high-level synthesis
Re: We thank the reviewer for this excellent suggestion. We agree that a concise visual summary of the epidemiological shifts in GAS phenotypes across the pre-antibiotic era, antibiotic era, and post-COVID-19 period would greatly enhance the clarity and accessibility of the review. In response, we have created a new timeline-style figure (Figure 1) that synthesizes major epidemiological milestones, changes in dominant GAS phenotypes and shifts in incidence patterns, including the impact of the COVID-19 pandemic (pp. 9-10).

Reviewer 3 Report

Comments and Suggestions for Authors

This manuscript addresses an important and timely topic with commendable clarity and depth. However, I would like to offer several suggestions aimed at enhancing the overall impact of the paper and increasing its relevance and appeal to a clinical readership:

  1. Introduction: It may strengthen the introduction to include a brief discussion of the differences in clinical presentation between immunocompromised and immunocompetent individuals. Highlighting these distinctions early on would contextualize the significance of diagnostic and therapeutic challenges across varying patient populations.

  2. Methodology: If this work is intended as a systematic review, it would benefit from the inclusion of a PRISMA (Preferred Reporting Items for Systematic Reviews and Meta-Analyses) flow diagram. This addition would improve transparency and methodological rigor by clearly delineating the study selection process. Furthermore, greater clarity is needed regarding the inclusion and exclusion criteria, particularly the rationale for excluding certain studies.

  3. Discussion – Epidemiology (Section 4): The epidemiological subsection is notably detailed; however, it currently appears disproportionately lengthy compared to the rest of the discussion. Condensing this section to focus on the most pertinent epidemiological trends and data would enhance readability and maintain the narrative flow.

  4. Future Outlook: The section outlining future directions could be strengthened by incorporating a discussion on emerging vaccine development efforts and novel antimicrobial strategies, especially in the context of rising antimicrobial resistance. This is particularly relevant for clinicians managing severe and rapidly progressing infections such as necrotizing soft tissue infections, where conventional therapies may be insufficient.

Implementing these revisions may enhance the manuscript’s clinical relevance and broaden its appeal to clinicians 

Author Response

This manuscript addresses an important and timely topic with commendable clarity and depth. However, I would like to offer several suggestions aimed at enhancing the overall impact of the paper and increasing its relevance and appeal to a clinical readership:
Re: Thank you for your suggestions. We revised the manuscript according to your comments.

  •    Introduction: It may strengthen the introduction to include a brief discussion of the differences in clinical presentation between immunocompromised and immunocompetent individuals. Highlighting these distinctions early on would contextualize the significance of diagnostic and therapeutic challenges across varying patient populations.
    Re: Added as recommended (p. 2).
  •     Methodology: If this work is intended as a systematic review, it would benefit from the inclusion of a PRISMA (Preferred Reporting Items for Systematic Reviews and Meta-Analyses) flow diagram. This addition would improve transparency and methodological rigor by clearly delineating the study selection process. Furthermore, greater clarity is needed regarding the inclusion and exclusion criteria, particularly the rationale for excluding certain studies.
    Re: It is clearly written that this manuscript is a narrative review. For this reason, PRISMA is not included. 
    3.    Discussion – Epidemiology (Section 4): The epidemiological subsection is notably detailed; however, it currently appears disproportionately lengthy compared to the rest of the discussion. Condensing this section to focus on the most pertinent epidemiological trends and data would enhance readability and maintain the narrative flow.
    Re: We revised the text according to your suggestions and those received from the other reviewers (pp. 6-10). We included a concise visual summary of the epidemiological shifts in GAS phenotypes across the pre-antibiotic era, antibiotic era, and post-COVID-19 period to enhance the clarity and accessibility of the review (p. 10).
    4.    Future Outlook: The section outlining future directions could be strengthened by incorporating a discussion on emerging vaccine development efforts and novel antimicrobial strategies, especially in the context of rising antimicrobial resistance. This is particularly relevant for clinicians managing severe and rapidly progressing infections such as necrotizing soft tissue infections, where conventional therapies may be insufficient.
    Re: We appreciate the reviewer’s insightful suggestion. In response, we have expanded the "Future Outlook" section to include a more detailed discussion of current progress in Group A Streptococcus (GAS) vaccine development, including multivalent M protein-based approaches, conserved antigen targets, and efforts to overcome historical challenges related to strain diversity and potential autoimmune cross-reactivity (p. 11). Additionally, we have incorporated a summary of novel antimicrobial strategies, such as phage-derived lysins, peptide antibiotics, and anti-virulence therapies, which may offer adjunctive or alternative treatment options in the face of escalating resistance—particularly for severe infections like necrotizing soft tissue infections, where rapid clinical deterioration can outpace standard therapy (p. 11). We believe this addition meaningfully enhances the clinical relevance of the review for frontline healthcare professionals and aligns with the urgent need for innovation in both prevention and treatment of invasive GAS infections.
  • Implementing these revisions may enhance the manuscript’s clinical relevance and broaden its appeal to clinicians.
    Re: Thank you for your suggestion. We revised the manuscript according to your suggestions and those received from the other reviewers.